**Data Availability Statement:** All data and codes for this project are uploaded in the following Github

# Comparative impact assessment of COVID-19 policy interventions in five South Asian countries using reported and estimated unreported death counts during 2020-2021

**Ritoban Kundu**[1]*, **Jyotishka Datta**[2], **Debashree Ray**[3], **Swapnil Mishra**[4], **Rupam Bhattacharyya**[1], **Lauren Zimmermann**[5], **Bhramar Mukherjee**[1,6,7]

1 Department of Biostatistics, University of Michigan, Ann Arbor, Michigan, United States of America, 2 Department of Statistics, Virginia Polytechnic Institute and State University, Blacksburg, Virginia, United States of America, 3 Department of Epidemiology, Johns Hopkins University, Baltimore, Maryland, United States of America, 4 School of Public Health National University of Singapore, Singapore, Singapore, 5 Department of Psychiatry, University of Michigan, Ann Arbor, Michigan, United States of America, 6 Biostatistics Unit, Medical Research Council, University of Cambridge, Cambridge, United Kingdom, 7 Department of Epidemiology, University of Michigan, Ann Arbor, Michigan, United States of America

* kundur@umich.edu

## Abstract

There has been raging discussion and debate around the quality of COVID death data in South Asia. According to WHO, of the 5.5 million reported COVID-19 deaths from 2020-2021, 0.57 million (10%) were contributed by five low and middle income countries (LMIC) countries in the Global South: India, Pakistan, Bangladesh, Sri Lanka and Nepal. However, a number of excess death estimates show that the actual death toll from COVID-19 is significantly higher than the reported number of deaths. For example, the IHME and WHO both project around 14.9 million total deaths, of which 4.5–5.5 million were attributed to these five countries in 2020-2021. We focus our gaze on the COVID-19 performance of these five countries where 23.5% of the world population lives in 2020 and 2021, via a counterfactual lens and ask, to what extent the mortality of one LMIC would have been affected if it adopted the pandemic policies of another, similar country? We use a Bayesian semi-mechanistic model developed by Mishra et al. (2021) to compare both the reported and estimated total death tolls by permuting the time-varying reproduction number ($R_t$) across these countries over a similar time period. Our analysis shows that, in the first half of 2021, mortality in India in terms of reported deaths could have been reduced to 96 and 102 deaths per million compared to actual 170 reported deaths per million had it adopted the policies of Nepal and Pakistan respectively. In terms of total deaths, India could have averted 481 and 466 deaths per million had it adopted the policies of Bangladesh and Pakistan. On the other hand, India had a lower number of reported COVID-19 deaths per million (48 deaths per million) and a lower estimated total deaths per million (80 deaths per million) in the second half of 2021, and LMICs other than Pakistan would have lower reported mortality had they followed India's strategy. The gap between the reported and estimated total deaths highlights the varying level and extent of under-reporting of deaths across the subcontinent, and that model

repository given by: https://github.com/Ritoban1/COVID-Counterfactual.

**Funding:** This project is supported by two grants of BM, namely NSF DMS 1712933 and NIH/NCI CA267907. The funders had no role in study design, data collection and analysis, decision to publish, or preparation of the manuscript.

**Competing interests:** The authors have declared that no competing interests exist.

estimates are contingent on accuracy of the death data. Our analysis shows the importance of timely public health intervention and vaccines for lowering mortality and the need for better coverage infrastructure for the death registration system in LMICs.

## Introduction

COVID-19 created a polycrisis all over the world and the impact has been felt everywhere, with almost no country spared. Low and middle-income countries (LMICs) initially faced a steep challenge in providing medical services with a fragile health infrastructure [1] and scarce social safety-nets. However, during the course of the pandemic, the overall performances of the LMICs in responding to the global crisis improved, and some LMICs, like Vietnam [2], were able to effectively contain the spread of virus through early and aggressive public health measures such as testing, tracing and isolating cases. In this article, we focus on South Asian LMICs, viz., India, Bangladesh, Pakistan, Sri Lanka and Nepal through a counterfactual lens of their relative mortality burdens via unreported deaths. Our study is motivated by three key factors. Firstly, as Younus (2021) [3] commented, the pandemic created a severe public health and economic crises pushing millions of households into poverty and substantially increasing income and wealth inequalities. Despite the huge literature on the impact of Covid-19 on these countries, there is essentially no prior work on comparing their relative performances that could provide a basis for policy comparison and build the ground for future prevention plans.

Secondly, with hybrid immunity against the virus acquired via the cumulative effect of natural and vaccine immunity, LMICs like India faced a less severe Omicron compared to the previous two waves [4]. For example, Babu et al. [5] analyzed the pandemic responses of eight South Asian countries and found that these countries had lower mortality rates from COVID-19 than many other countries worldwide, making counterfactual comparisons between UK and European countries [6] less informative in the context of SA LMICs. Finally, as we discuss below, while there are pronounced similarities in certain aspects of their social determinants of health, existing comorbidities and the demographic indicators of the five countries (vide Fig 1), there are also marked variations in these key indicators, and this duality underscores the importance of the relative comparison of pandemic performance while recognizing the contextual differences. We would also like to note here that the comparison of these pre-pandemic metrics only serve as a rationale for this study and not used as evidence or input in our methodology.

A key challenge in modeling pandemic trajectory in LMICs is the significantly lower data resolution and quality compared to high-income countries, as highlighted by prior research [7]. This disparity in data quality contributes to the challenge of under-reporting of deaths in LMICs, thereby complicating policy evaluation in these countries [8]. As such, assessing data quality and developing reliable estimates of total infections and fatalities are crucial prerequisites for modeling and conducting counterfactual evaluations in LMICs. It is recommended that total mortality estimates be used in these evaluations to address issues of selection bias and under-reporting [9, 10]. Any fair comparison across LMIC countries should consider death underreporting instead of relying on the reported number of deaths. Several such estimates for excess deaths during the pandemic are now available [11–13].

Mishra et al. [6] explored using a semi- mechanistic Bayesian model how timing and effectiveness of control measures in the UK, Sweden and Denmark shaped COVID-19 mortality in each country using a counterfactual assessment: *what would the impact have been, had each country adopted the others' policies?* Their results show that small changes in the timing or

| Metric | India | Bangladesh | Pakistan | Nepal | Sri Lanka |
|---|---|---|---|---|---|
| Hypertension (Adults) | 207 (21.6%) (2015-16) [17] | 18.1 (15.1%) (2010) [18] | 38.7 (27.8%) (2013) (WHO) | 4.7 (24.8%) (2019) (WHO) | 3.6 (21.9%) (2014) (WHO) |
| Population with COPD | 55.3 (5.8%) (2016) [19] | 14.5 (12.1%) (2016) [20] | 9.3 (6.6%) (2016) [21] | 2.3 (12.1%) (2017) [22] | 1.46 (8.9%) (2017) [23] |
| Population with asthma | 37.9 (3.9%) (2020) [24] | 6.9 (5.8%) (2013) [25] | 9.31 (6.7%) (2015) [26] | 2.59 (13.7%) (2019) [27] | 2.59 (15.8%) (2016) [28] |
| Population with diabetes (adult) | 139.3* (14.5%) (2017) | 13.8* (11.5%) (2017) | 18.08* (13.0%) (2021) | 3.11* (16.4%) (2021) | 1.59* (9.7%) (2021) |
| Population (0-14 yrs) | 383.5 (28.6%) ** (2021) | 44.8 (27.2%)** (2021) | 77.3 (35.7%)** (2021) | 10.5 (34.9%)** (2021) | 5.5 (25.2%)** (2021) |
| Population (15-64 yrs) | 852.9 (63.6%)** (2021) | 111.4 (67.6%)** (2021) | 130.4 (60.2%)** (2021) | 15.8 (59.8%)** (2021) | 14.7 (66.9%)** (2021) |
| Population (65+ yrs) | 71.0 (5.3%)** (2021) | 8.5 (5.2%)** (2021) | 8.9 (4.1%)** (2021) | 1.4 (5.3%)** (2021) | 1.7 (7.8%)** (2021) |
| Hospital Beds | 0.71* (0.05%) (2010) | 0.13* (0.08%) (2010) | 0.13* (0.06%) (2010) | 0.11* (0.38%) (2010) | 0.08* (0.36%) (2010) |
| Cardiovascular deaths | 3.8* (0.4%) (2017) | 0.49* (0.41%) (2017) | 0.92* (0.66%) (2017) | 0.06* (0.32%) (2017) | 0.04* (0.24%) (2017) |
| GDP (2019) (USD) | 2101*** | 1856*** | 1482*** | 1195*** | 4077*** |
| GDP (2020) (USD) | 1901*** | 1969*** | 1194*** | 1155*** | 3682*** |

**Fig 1. Number of people along with their percentages for different comorbidities.** This includes hypertension (Adults) [17, 18], COPD [19–23], asthma [24–28], diabetes (Adults), cardiovascular death rate, no of hospital beds, and number of people in different age groups and GDP per capita (2019 and 2020) for each of the five countries. Population in each category reported in millions. The percentages for comorbidities are calculated within population at risk.*ourworldindata.org. **wikipedia.org, ***worldbank.org.

effectiveness of interventions have disproportionately large effects on reported mortality due to COVID-19 pandemic. We extend the work of Mishra et al. [6] by including an in-depth evaluation of five LMICs from South Asia, namely India, Bangladesh, Pakistan, Sri Lanka and Nepal in 2020–2021. We choose to focus on these five countries as they are home to 23.5% of the population of the world and more importantly, there exists no prior across-country 'what if?' analysis for these LMICs. Furthermore, the critical issue of under-reporting of deaths was accounted for in our analysis by using the IHME (The Institute for Health Metrics and Evaluation) excess mortality estimates. We compare and contrast the counterfactual results from both reported and estimated total mortality. The analysis of total estimated deaths using a counterfactual framework is a novel contribution of this paper. This has been a first attempt to systematically compare the five South Asian countries of interest in terms of both reported and estimated total deaths.

Although we cannot comprehensively account for all the differences within and between countries, our analysis provides a starting point for performing counterfactual analysis in the LMIC context. This is further substantiated by Salvatore et al. [14], who consider similar analysis but only for India generating counterfactuals for policy interventions, but no work on pair-wise counterfactuals across LMICs are available till date. These countries under consideration are geographically close to each other and share common social determinants of health (SDOH) such as, poverty, inequality, rapid urbanization, unhygienic living conditions, strong child immunization programs, healthcare accessibility issues, and challenges with disease surveillance and public health systems [15, 16]. Our approach using total estimated deaths tries to place all countries on the same playing field and control for differences in reporting COVID deaths.

To set the backdrop, a pre-pandemic summary of these countries' socio-economic and demographic relevant factors for COVID-19 is presented in Fig 1. The details are discussed in the Results Section. Despite the variations in GDP, comorbidities, and age distributions presented in Fig 1, there are similarities in culture, geography, poverty levels, and healthcare

quality among the five countries of interest, making a quasi-experimental counterfactual assessment of their COVID-19 situations valuable from a *global public health* perspective. In terms of reported COVID-19 deaths, about 70% of the deaths from 2020 till 2021 occurred in 2021 for these five South Asian countries of interest. Our main focus of interest in this study is assessing the COVID-19 pandemic situation in 2021 in terms of reported and estimated total deaths. We also consider the 2020 data separately as some of the policy measures were more centrally coordinated and stringent in 2020, the variant and the immunity landscape was different from 2021. A detailed description of the different interventions implemented by the five South Asian countries of interest in 2020 and 2021 are provided in S8 and S9 Tables respectively. The summary synthesis of these tables are presented in the Results Section. This table serves as an important archive of the policy interventions implemented by these countries since several of the sources of information are not publicly available anymore. The measures undertaken by the different governments were contingent upon the distinct circumstances, socio-political influences and health infrastructure in each country, leading to an ensemble of divergent approaches in managing the pandemic situation. Hence, a comprehensive counterfactual analysis to assess the impact of the policy measures undertaken in these five countries is of high significance.

## Methods

### Data aggregation

The COVID-19 reported death data for the five South Asian countries, namely India, Bangladesh, Pakistan, Sri Lanka, and Nepal, were collected from the JHU CSSE COVID database. The study utilizes IHME estimates from covid19.healthdata.org for total deaths, that accounts for under reporting. Mobility Data were obtained from the Google mobility database using various metrics. We report mobility changes for transit stations, retail and recreation, grocery and pharmacy, parks, workplaces, and residential areas. A detailed description of the different sources along with relevant website links are provided in S4 Table.

### Models

We used a Bayesian semi-mechanistic renewal process model for both reported and estimated total deaths for all five countries. Fig 2 describes a summary of the specifications of the stochastic components of the model. One important point to note is that the renewal equations in our work do not use the daily reported cases as input data to estimate the model parameters but rather use the daily reported and total estimated deaths from IHME as the only input data. Infections (not reported cases) are inferred from the death data, the infection fatality rate (IFR) and the model for country specific time varying reproduction number using the renewal equation.

**Transmission model.** *Observed data.* $D_{t,m}$—Reported number of deaths attributed to COVID-19, where $t$ denotes the day or time and $m \in \{1, 2, 3, 4, 5\}$ denotes the country.

$D^*_{t,m}$—Total number of estimated deaths from IHME (sum of reported and unreported) attributed to COVID-19, where $t$ denotes the day or time and $m \in \{1, 2, 3, 4, 5\}$ denotes the country. However, one can use estimates from any source.

*Likelihood.* Let $d_{t,m} = \mathbb{E}(D_{t,m})$ denote the expected number of reported deaths. The distributional assumptions for $D_{t,m}$ used here to define the likelihood is given by

$$D_{t,m} \sim \text{Negative Binomial}\left(d_{t,m}, d_{t,m} + \frac{d^2_{t,m}}{\psi}\right)$$

$$\psi \sim \mathcal{N}^+(0, 5)$$

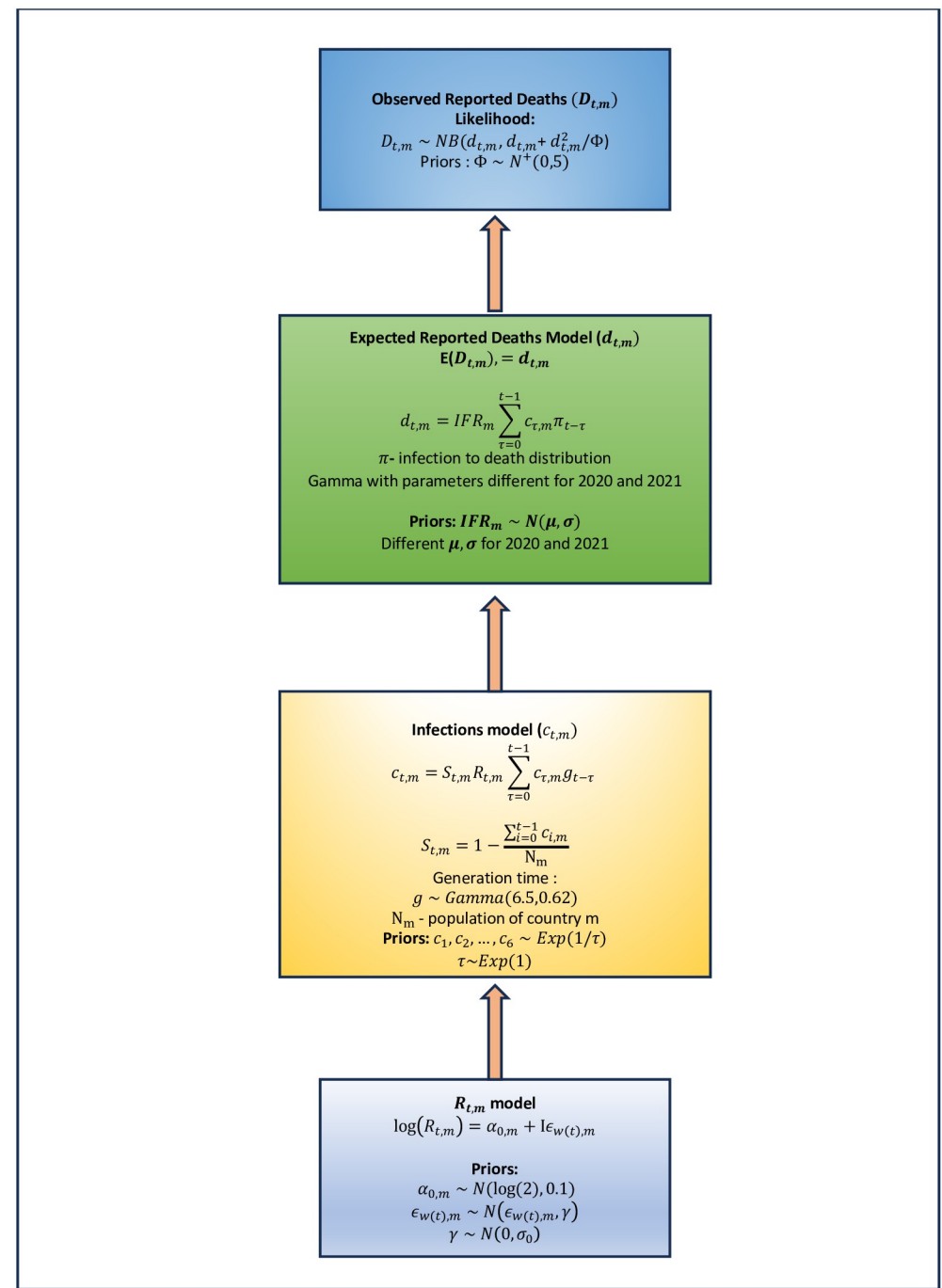

**Fig 2. Schematic representation of the different components of the Bayesian semi- mechanistic transmission model.** Each box represents and provides details of the components that define the overall stochastic structure of the model.

Similarly for estimated total mortality analysis we defined $d_{t,m}^* = \mathbb{E}(D_{t,m}^*)$ to denote the expected number of total (reported and unreported) estimated deaths. The distributional assumptions for $D_{t,m}^*$ used here to define the likelihood is given by

$$D_{t,m}^* \sim \text{Negative Binomial}\left( d_{t,m}^*, d_{t,m}^* + \frac{d_{t,m}^{*2}}{\psi} \right)$$

$$\psi \sim \mathcal{N}^+(0,5)$$

where, $\mathcal{N}^+(\mu, \sigma)$ is a half-normal distribution with mean $\mu$ and standard deviation $\sigma$. $X \sim \mathcal{N}^+(\mu, \sigma)$ if, $X \sim |Y|$ and $Y \sim \mathcal{N}(\mu, \sigma)$. Note that one limitation is that we consider the point estimates $D_{t,m}^*$ obtained from IHME as the estimated total deaths and ignore the uncertainty associated.

**Renewal equation.** *Linking mortality to infections*. The relation between the observed expected number of deaths, $d_{t,m}$ on a given day $t$ for the country $m$ with the past number of daily infections $c_{\tau,m}$ (observed + unobserved) is:

$$d_{t,m} = \text{IFR}_{\text{Rep,m}} \sum_{\tau=0}^{t-1} c_{\tau,m} \pi_{t-\tau}, \tag{1}$$

where $\text{IFR}_{\text{Rep,m}}$ denotes the infection fatality rate for the reported deaths analysis defined as probability of a reported death given infection.

On the other hand, the relation between the observed expected number of total estimated deaths, $d_{t,m}^*$ on a given day $t$ for the country $m$ with the past true number of daily infections $c_{\tau,m}$ (observed + unobserved) is:

$$d_{t,m}^* = \text{IFR}_{\text{Tot,m}} \sum_{\tau=0}^{t-1} c_{\tau,m} \pi_{t-\tau}, \tag{2}$$

where $\text{IFR}_{\text{Tot,m}}$ denotes the infection fatality rate for the total estimated deaths analysis defined as probability of total estimated death (reported and unreported) given infection.

Here $\pi$ is the discretized density of the infection to death distribution. Let us concentrate on Eq (1) for the time being. The intuition behind Eq (2) is similar. The expression $\text{IFR}_{\text{Rep,m}} \cdot c_{\tau,m} \pi_{t-\tau}$ denotes the expected reported number of patients to die on day $t$ for the country $m$ who developed their infection on day $\tau$. Therefore $\text{IFR}_{\text{Rep,m}} \sum_{\tau=0}^{t-1} c_{\tau,m} \pi_{t-\tau}$ denotes the number of expected deaths on day $t$.

*Linking current infections to past infections using reproduction number*. The true number of infected individuals is modeled using a discrete renewal process and related to the past infected counts using the following equation:

$$c_{t,m} = S_{t,m} R_{t,m} \sum_{\tau=0}^{t-1} c_{\tau,m} g_{t-\tau} \tag{3}$$

$$S_{t,m} = 1 - \frac{\sum_{i=1}^{t-1} c_{i,m}}{N_m} \tag{4}$$

where $g$ denotes the discretized version of the generation time density. The population size of country $m$ is specified by $N_m$. The infections at day $t$ depend on the number of infections in previous days weighted by the discretized generation time distribution. This weighting is then multiplied by the country specific time varying reproduction number $R_{t,m}$ which denotes the

average number of secondary infections per infection at a given time. The adjustment factor $S_{t,m}$ accounts for the depletion of susceptibles in the population because even in absence of interventions herd immunity will reduce the number of daily infections through susceptible populations. One important point is that $R_{t,m}$ is unadjusted for population depletion, however the term $S_{t,m}R_{t,m}$ takes into account the adjusted reproduction number for population depletion. Without including the depletion factor into $R_{t,m}$ helps to considerably improve posterior topology and inference. However while generating the counterfactuals, the adjustment factor is always included to account for susceptible depletion.

*Structure of the model for $R_{t,m}$.* $R_{t,m}$ for day $t$ and country $m$ is defined as the country specific time varying reproduction number which models the average number of secondary infections per infection at a given time. In our main analysis, the unadjusted $R_{t,m}$ for day $t$ and country $m$ is formulated as a random process

$$\log(R_{t,m}) = \alpha_{0,m} + \mathbb{I}\epsilon_{w(t),m} \tag{5}$$

where the conversion from days to weeks is encoded in $w(t)$. Therefore $R_{0,m} = e^{\alpha_{0,m}}$. Every 7 days, $w$ is incremented, i.e we set $w(t) = [(t - t_{\text{start}})/7] + 1$, where $t_{\text{start}}$ is the date of intervention. For all $t < t_{\text{start}}$, $\epsilon_{w(t),m}$ is set to zero. For $t \geq t_{\text{start}}$, to specify the random walk process, we introduce the parameter $\gamma \sim \mathcal{N}(0, \sigma_0)$ and then model $\epsilon$ as $\epsilon_{w(t),m} \sim \mathcal{N}(\epsilon_{w(t)-1,m}, \gamma)$. We model $\log(R_{t,m})$ to ensure positivity of $R_{t,m}$. The indicator function is used to ensure that the random walk can only start at the date of intervention which is $t_{\text{start}}$.

**Specification/choice of priors and other input parameters.** *Prior specification on the Infection Fatality Rate (IFR).* A major challenge exists in obtaining realistic values of the prior distribution parameters for IFR(s) in four of the countries we consider (Pakistan, Nepal, Sri Lanka and Bangladesh). These four countries lack reliable serosurveys or linelist (cases/deaths) data. We tackle this challenge by using the IFR estimates of India based on serosurveys and assuming that the ratio of IFRs between two countries is the same as that of Case Fatality Ratios (CFR) in those two countries. Case fatality Ratio (CFR) for a country at day $t$ is defined as the ratio of cumulative number of reported deaths and cumulative number of reported cases till day $t$. More formally, for any two countries $x$ and $y$,

$$\frac{\text{IFR}_x}{\text{IFR}_y} = \frac{\text{CFR}_x}{\text{CFR}_y}. \tag{6}$$

where IFR can be both $\text{IFR}_{\text{Rep}}$ and $\text{IFR}_{\text{Tot}}$. The IFRs for India in 2020 and 2021 were derived based on the serial serosurvey estimates in Zimmermann et al. [8]. The CFR for all of the countries can be obtained using the JHU CSSE COVID database. Based on these serosurvey estimates for India and using the assumption that the ratio of IFRs are same as that of CFR, we obtained the estimates of mean and standard deviation of the prior distributions of IFRs in other countries that did not have serial serosurveys. Specifically, in 2020 we assumed the prior distributions of $\text{IFR}_{\text{Rep},m}$ and $\text{IFR}_{\text{Tot},m}$ (Infection fatality rates for reported and estimated total mortality analysis respectively) to be normal distributions with (mean, sd) as (0.05,0.03) and (0.37,0.21) respectively. In contrast, in 2021 we assumed the prior distribution of $\text{IFR}_{\text{Rep},m}$ and $\text{IFR}_{\text{Tot},m}$ follows normal distributions with (mean, sd) as (0.08,0.05) and (0.39,0.23) respectively. The means and standard deviations of the IFRs are chosen based on the obtained IFRs of all the five countries. However, within a given year of analysis, we adopted a fixed prior for Infection Fatality Rate (IFR) to ensure the identifiability of the parameters in Eqs (1) and (2).

*Infection to death distribution.* In our study, we modeled the infection-to-death distribution ($\pi$) in Eqs (1) and (2) as the sum of two independent distributions: the incubation period (time from infection to the onset of symptoms) and the time between the onset of symptoms and

deaths. To capture variations in the virus strain, we considered two different incubation period distributions based on the specific time period under investigation. For instance, in the year 2020, the mean incubation period of the ancestral COVID variant was set at 5 days based on existing literature [29]. Consequently, we formulated the incubation period as a random variable following the Gamma distribution with shape and scale parameters of 5.8 days and 0.94, respectively. We used the same onset to death distribution (Gamma distribution with shape and scale parameters 1.45 days and 10.43 respectively [6] for 2020 and 2021. Therefore in 2020, the infection to death distribution is the sum of two independent Gamma variables given by

$$\pi \sim \text{Gamma}(5.8, 0.9) + \text{Gamma}(1.45, 10.43)$$

In contrast, in 2021 with delta strain as the major variant of COVID, we used an incubation period with a distribution of mean 4 days [95% CI (3,5)] [30]. Therefore the distributions of the incubation period differ across 2020 and 2021 in our analysis. In 2021, the infection to death distribution is the sum of two independent Gamma variables given by

$$\pi \sim \text{Gamma}(62.22, 0.06) + \text{Gamma}(1.45, 10.43)$$

The discrete probability mass function (pmf) $\pi_t$ for day $t$ was obtained using the expression, $\pi_t = \int_{t-0.5}^{t+0.5} \pi(\tau)d\tau$ for $t = 0, 1, 2, 3,..$

*Generation time*. Generation time is defined as the time between when a person becomes infected and when they subsequently infect another person. Usually the generation time is unknown. Similar to [6], we approximate the generation time distribution $g$ with density $g(\tau)$ by the serial interval distribution (time from symptom onset in one person to the time of symptom onset in the person they infect). We chose both the generation time and the serial interval to be Gamma distribution with mean 6.5 days and standard deviation 4.2 [31], where authors used the line-list data from China during the start of the pandemic. We would like to emphasize that we have not used the line-list data from countries of interest in the paper. The access to line-list data is not available to us. Moreover in these countries there was absence of any effective contact tracing to really estimate the generation distribution from the line-list data. The generation time distribution is discretized by $g_t$ for day $t$ was obtained using the expression $g_t = \int_{t-0.5}^{t+0.5} g(\tau)d\tau$, $t = 2, 3, \ldots$ and $g_1 = \int_0^{1.5} g(\tau)d\tau$.

*Priors on the parameters of the model for $R_{t,m}$ and choice of the seeding parameters*. For 2020, we used $t_{\text{start}}$ to be March 15, 2020. On the other hand, $t_{\text{start}}$ for the 1st and 2nd halves of 2021 were given by January 1, 2021 and July 1, 2021. Since the basic reproduction number is given by $R_{0.m} = e^{\alpha_{0,m}}$, the prior distribution of $\alpha_{0,m}$ was chosen to be $\mathcal{N}(\log(2), 0.1)$ which can represent a broad range of plausible mean basic reproduction numbers with support from 1 to 3 (obtained by the `EpiEstim` package in `R` for the five countries of interest in 2020 and 2021). We seeded the model with six sequential days of an equal number of infections given by $c_{1,m} = c_{2,m} = \ldots = c_{6,m} \sim \text{Exponential}\left(\frac{1}{\tau}\right)$. In S1 Text, we discuss in details the estimation of true number of infections $c_{t,m}$ and the choice of $\tau$ used. The transmission model parameters are estimated in the probabilistic programming language Stan [32] using an adaptive Hamiltonian Monte HMC sampler.

*Constructing counterfactual scenarios*. We follow the exact structure for creating the counterfactual model as detailed in [6]. $R_{0,m}$ for country $m$ is defined as the average number of secondary infections produced by a typical case of an infection in a population where everyone is susceptible. In contrast, $R_{t,m}$ for day $t$ and country $m$ is defined as the country specific time varying reproduction number which models the average number of secondary infections per infection at a given time. Under this setting, a country's time-varying reproduction number

                                        

$(R_{t,m})$ is replaced by the $R_{t,m}$ of a different country. Then, the new resulting $R_{t,m}$ is used to simulate the evolving epidemics for that country. For the remainder of the text, we refer to the country from which the intervention scenario is considered for constructing a counterfactual situation as the donor, and the country whose mortality data under the donated intervention is being analyzed as the recipient. We apply the relative reductions to the donor countries $R_{t,m}$ starting from the intervention date for generating the counterfactual. Denoting $x$ as the donor country and $y$ as the recipient country, the $R_{t,y}$ becomes $\frac{R_{0,y}}{R_{0,x}} R_{t,x}$ after the intervention date. The relative approach ensures that the counterfactual decrease is proportional to the initial dynamics of the recipient country. Mathematically, for two countries $x$ and $y$, the time varying reproduction number $R_t$ are modeled

$$R_{t,x} = e^{\alpha_{0,x} + \mathbb{I}\epsilon_{w(t),x}}$$
$$R_{t,y} = e^{\alpha_{0,y} + \mathbb{I}\epsilon_{w(t),y}}$$

Using the above mentioned relative approach in the counterfactual scenario where $x$ and $y$ are the donor and the recipient countries respectively, we have

$$R_{t,x \to y} = \begin{cases} e^{\alpha_{0,x} + \mathbb{I}\epsilon_{w(t),x}} & \text{if } t < t_{\text{start}} \\ \dfrac{e^{\alpha_{0,x}}}{e^{\alpha_{0,y}}} e^{\alpha_{0,y} + \mathbb{I}\epsilon_{w(t),y}} & \text{if } t \geq t_{\text{start}} \end{cases}$$

After cancelling the terms, we finally obtain the final expression for $R_{t,y \to x}$ as,

$$R_{t,x \to y} \begin{cases} e^{\alpha_{0,x} + \mathbb{I}\epsilon_{w(t),x}} & \text{if } t < t_{\text{start}} \\ e^{\alpha_{0,x} + \mathbb{I}\epsilon_{w(t),y}} & \text{if } t \geq t_{\text{start}} \end{cases}$$

Using Eqs (3) and (4), $R_{t_{\text{start}}, x \to y}$ and fitted true infections for $\tau < t_{\text{start}}$, we obtain estimates of $c_{t,x}$ for all $t \geq t_{\text{start}}$. Finally the estimates of $d_{t,x}$ and $d_{t,x}^*$ are simulated for the counterfactual situations putting the estimates of $c_{\tau,x}$ in Eqs (1) and (2) respectively.

**Time period of analysis.** *2020*. Due to extremely low reported and total estimated death counts, we excluded Sri Lanka from our analysis in 2020. Both the models specified by Eqs (1) and (2) were fitted from February 1, 2020 to December 31, 2020. The intervention date was chosen as March 15, 2020.

*2021*. The entire year of 2021 divided into two equal halves, namely January 1, 2021–June 30, 2021 and July 1, 2021–December 31, 2021. For these two halves both the models specified by Eqs (1) and (2) were fitted from November 15, 2020 and April 1, 2021 respectively. The interventions dates from these two halves were chosen as January 1, 2021 and July 1, 2021 respectively. We selected the start dates of model fitting based on model convergence in each respective period. The two halves are treated as discrete entities rather than a continuous stream of time.

**Sensitivity analysis.** We performed sensitivity analyses to assess the variation of the counterfactual results for different choices of parameter specifications of IFR distribution and generation time distribution. Additionally, we performed a sensitivity analysis to infer how different the counterfactual results will be affected by including mobility as a predictor of $R_{t,m}$. We defined $\text{mob}_{t,m}$ as the sum of the six types of variables available from mobility data at day $t$ for country $m$, where each of these variables have been standardized with respective means and standard deviations. The original stochastic model for the time varying reproduction number $R_{t,m}$ as in Eq (5) is modified as follows

$$\log(R_{t,m}) = \alpha_{0,m} + \mathbb{I}\epsilon_{w(t),m} + \beta \text{mob}_{t,m}$$

One can also use country specific $\beta_m$ in the above equation. Before fitting the above model we standardized the mobility data with respective mean and standard deviation for each country. The results of these analyses are presented in the Results section. All the sensitivity analyses have been performed using reported mortality data in the time period of Jan 1, 2021–June 30, 2021.

## Results

### Variability in socio-economic and demographic factors relevant for COVID-19 across the five countries

The prevalence of comorbidities associated with COVID-19 varied significantly across the five south Asian countries. A pre-pandemic summary of their socio-economic and demographic factors relevant for COVID-19 is presented in Fig 1. Sri Lanka had the highest percentage of the population with asthma (15.8%) and India the lowest (3.9%). The percentage of people with diabetes ranges from 9.7% in Sri Lanka to 16.4% in Nepal. However, some commonalities exist among all or subsets of these countries. For instance, India had only 0.71 hospital beds per million, while the others had around 0.10, which made the high number of severe symptomatic COVID-19 cases more of a significant concern for these countries. Furthermore, in Pakistan and Nepal, a high proportion of the population comprised young people aged 0–14 years, with percentages of 35.7% and 34.9%, respectively. India, Bangladesh, and Nepal have similar proportions of elderly people (65+), around 5.3%. Despite the variations in GDP, comorbidities, and age distributions, there are similarities in culture, geography, poverty levels, and healthcare quality among the five countries of interest, making a quasi-experimental counterfactual assessment of their COVID-19 situations valuable from a global public health perspective.

### Results in 2020

**Variability in COVID-19-related government interventions, lockdowns and vaccination initiatives across the five countries.** In 2020, India imposed four phases of nationwide lockdowns starting from March 25 to May 31 with dynamically adaptive measures to contain the spread of the virus (S8 Table). Starting from June 1, 2020 Indian government started to relax the restrictions with the six different unlock periods. Bangladesh observed a nationwide lockdown on similar time periods (March 26–May 27). In contrast, countrywide lockdown measures were imposed for a short span of time in Pakistan (April 1–May 9) compared to India or Bangladesh. Nepal experienced the longest period of lockdown which contributed towards lower COVID-19 mortality counts in the first half of 2020 (Fig 3). Moreover each country differed in terms of varying restrictions imposed during their respective lockdown period.

**Variability in observed pandemic-related metrics across countries.** For 2020, S1 Table and S1 Fig show that the mean daily $R_t$ for all the 5 countries were close to 1 and comparably similar at the start of the time period of analysis. The mean Case Fatality Rate (CFR) for Sri Lanka was lower than the other countries, while the test positive rates (TPR) for India and Pakistan were less than that of the other 3 countries, with Nepal and Bangladesh having the highest mean TPR of 0.13. From Figs 3 and 4 we observe that India experienced the highest number of reported deaths per million (77) in 2020 compared to the other countries. In contrast Nepal has the highest number of estimated total (reported and unreported) deaths per million (763) as observed from Figs 3 and 5. The estimated total deaths per million for India (725) and Pakistan (708) are comparable. We observe a decline in terms of both reported and total deaths due to COVID for India starting November to the end of 2020. S5 Fig shows the

## Reported and Total Deaths Per Million

**Fig 3. Daily reported and total (Reported + Unreported) deaths per million in 2020 (March 15 2020–Dec 31 2020) for each country.** The red and the black lines correspond to the daily reported and total estimated death trajectories respectively.

| Donor \ Recipient | India | Bangladesh | Pakistan | Nepal |
|---|---|---|---|---|
| **India** | 77 | Counterfactual : 147 [21,528]<br>Actual : 38<br>Averted : -109 [-490,17] | Counterfactual : 137 [17,486]<br>Actual : 37<br>Averted : -100 [-449,20] | Counterfactual : 74 [3,440]<br>Actual : 53<br>Averted : -21 [-387,50] |
| **Bangladesh** | Counterfactual : 12 [1,89]<br>Actual : 77<br>Averted : 65 [-12,76] | 38 | Counterfactual : 32 [5,205]<br>Actual : 37<br>Averted : 5 [-168,34] | Counterfactual : 15 [0,162]<br>Actual : 53<br>Averted : 52 [-110,52] |
| **Pakistan** | Counterfactual : 13 [1,101]<br>Actual : 77<br>Averted : 64 [-24,75] | Counterfactual : 39 [3,256]<br>Actual : 38<br>Averted : -1 [-218,34] | 37 | Counterfactual : 19 [0,209]<br>Actual : 53<br>Averted : 34 [-155,52] |
| **Nepal** | Counterfactual : 78 [4,540]<br>Actual : 77<br>Averted : -1 [-463, 72] | Counterfactual : 139 [10,740]<br>Actual : 38<br>Averted : -101 [-701,28] | Counterfactual : 135 [9,708]<br>Actual : 37<br>Averted : -98 [-670,28] | 53 |

**Fig 4. Reported cumulative deaths per million with the 95% CrI on December 31, 2020 calculated starting from March 15, 2020 in different counterfactual situations.** The rows correspond to the countries which are the donors, while the columns correspond to the recipients. The diagonal cells denote the reported deaths per million in the 5 countries. The off-diagonal cells correspond to the counterfactuals, which consists of 3 quantities: Counterfactual Deaths Per Million, Actual Deaths per million and difference of deaths between the actual and counterfactuals that could have been averted.

percent change from baseline for all the types of mobilities other than Residential is high for Pakistan. In the four categories other than Grocery-Pharmacy and Residential, the percent changes in mobility were moderate for India and for Nepal, in all of the cases, it was either on the lower or moderate side. For Sri Lanka other than Residency, there was a similar trend of

| Donor \ Recipient | India | Bangladesh | Pakistan | Nepal |
|---|---|---|---|---|
| **India** | 725 | Counterfactual : 869 [437,7886]<br>Actual : 509<br>Averted : -360 [-7377,72] | Counterfactual : 1420 [909,2347]<br>Actual : 708<br>Averted : -712 [-1639,-201] | Counterfactual : 2 [0,51]<br>Actual : 763<br>Averted : 761 [712,763] |
| **Bangladesh** | Counterfactual : 583 [13,1350]<br>Actual : 725<br>Averted : 142 [-625,712] | 509 | Counterfactual : 857 [89,1388]<br>Actual : 708<br>Averted : -149 [-680,619] | Counterfactual : 1 [0,11]<br>Actual : 763<br>Averted : 762 [752,763] |
| **Pakistan** | Counterfactual : 877 [365,1829]<br>Actual : 725<br>Averted : -151 [-1103,361] | Counterfactual : 444 [242,10542]<br>Actual : 509<br>Averted : 65 [-10033,267] | 708 | Counterfactual : 1 [0,34]<br>Actual : 763<br>Averted : 762 [729,763] |
| **Nepal** | Counterfactual : 10360 [281,31973]<br>Actual : 725<br>Averted : -9635 [-31247, 444] | Counterfactual : 9201 [1303,34934]<br>Actual : 509<br>Averted : -8692 [-34425,-794] | Counterfactual : 10267 [931,30463]<br>Actual : 708<br>Averted : -9559 [-29755,-222] | 763 |

**Fig 5. Total cumulative deaths (reported and unreported) with the 95% CrI on December 31, 2020 calculated starting from March 15, 2020 in different counterfactual situations.** The rows correspond to the countries which are the donors, while the columns correspond to the recipients. The diagonal cells denote the estimated actual deaths per million in the 4 countries. The off diagonal cells correspond to the counterfactuals, which consists of 3 quantities: Counterfactual Deaths Per Million, Actual Deaths per million, and difference of deaths between the actual and counterfactuals that could have been averted. A cell is coloured red or blue depending upon whether the 95% CrI of the number deaths averted in a counterfactual situation lies entirely in the negative or positive side.

rise till September, 2020 and then there was a decline. We exclude Sri Lanka from the 2020 counterfactual analysis due to the small number of deaths (both reported and total) and cases in 2020 as observed from Fig 3.

**Reported deaths analysis.** From Figs 4 and 6 we observe that in terms of our counterfactual analysis based on just reported deaths India could have averted a substantial number of fatalities in all counterfactual situations where India is the recipient except the scenario Nepal → India. In terms of the counterfactual scenarios Bangladesh → Pakistan (CrI Averted 5 [-168,34]) and Pakistan → Bangladesh (CrI Averted -1 [-218,34]), we observe that both these countries are comparable in terms of pandemic performance. In contrast Nepal could have averted a considerable number of deaths in counterfactual scenarios with Bangladesh and Pakistan as donors (95% $\text{CrI}_{Bangladesh \to Nepal}$ 38 [-110,52] and 95% $\text{CrI}_{Pakistan \to Nepal}$ 34 [-155,52]). Overall in terms of reported mortality in 2020, Pakistan and Bangladesh were ahead in terms of pandemic control followed by Nepal and India.

**Total estimated deaths analysis.** From Fig 5 in terms of estimated total (reported and unreported) deaths, we observe that Nepal could have averted a substantial number of total deaths per million (at least 761 [712, 763]) in all the three counterfactual scenarios with Nepal as the recipient. In terms of total deaths Bangladesh has the lowest estimated total deaths per million (509) compared to the other three countries. The pandemic performances of Bangladesh, Pakistan and India are comparable in terms of counterfactual scenarios. We observe that the relations are not symmetric. For example, the counterfactual scenarios (95% $\text{CrI}_{India \to Pakistan}$ -712 [-1639,-201] and 95% $\text{CrI}_{Pakistan \to India}$ -151 [-1103,361]) substantiate the inference

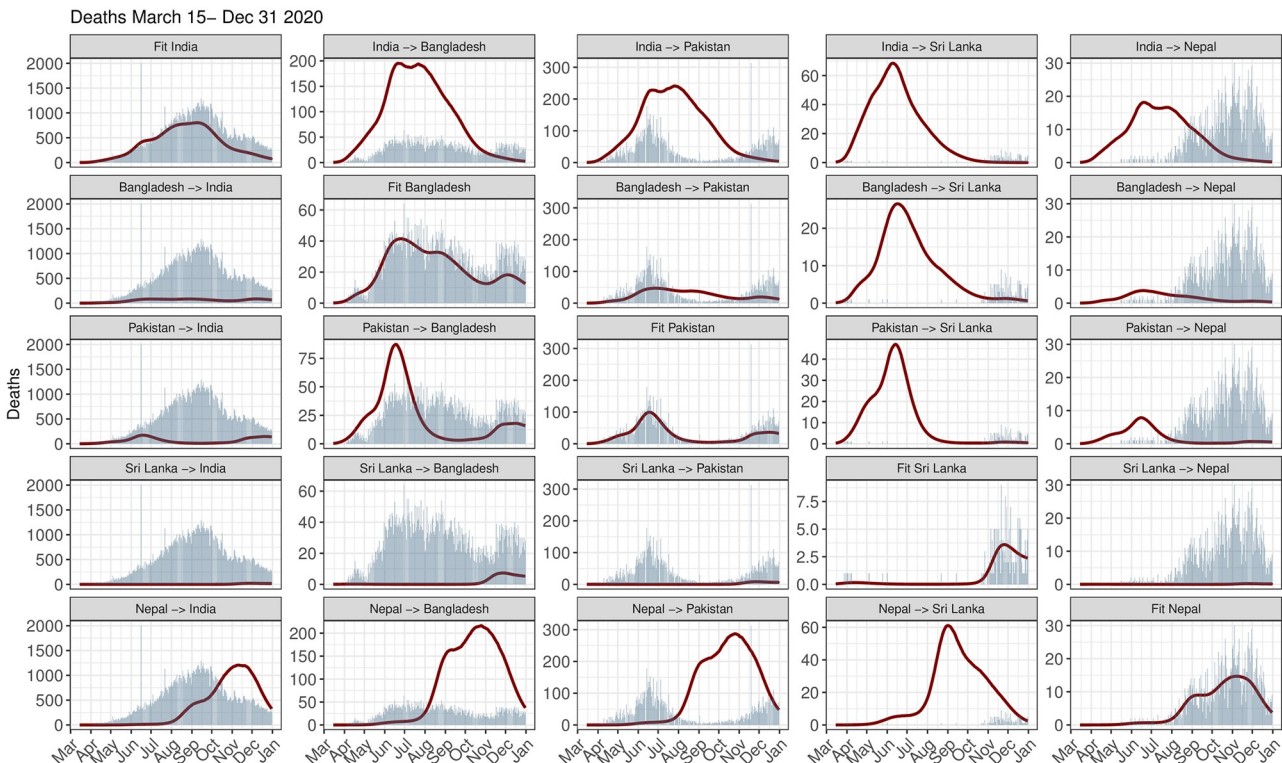

**Fig 6. Daily deaths for the different counterfactual situations along with the actual fitted reported deaths for each country in 2020 based on reported deaths analysis.** The blue bars in the plots denote the actual daily death cases for the recipient country, while the red lines denote the counterfactual ones. The time period of analysis is from March 15, 2020 to December 31, 2020.

that both nations would have incurred losses had they implemented the policy of the other country, indicating the uncertainty in the resultant estimates. In the context of total mortality in 2020, Bangladesh, Pakistan, and India demonstrate similar trends in pandemic control, while Nepal appears to fare the worst.

## Results in 2021

**Variability in COVID-19-related government interventions, lockdowns and vaccination initiatives across the five countries.** The initiation of the vaccination campaigns in these countries took place approximately between the late January and early February in the year of 2021 (S9 Table). With the exception of India, all four countries undertook comprehensive nationwide lockdowns for multiple days during the initial half of 2021. There were disparities in the length and commencement of these lockdowns across the countries. Conversely, India implemented region-specific lockdowns within individual states as opposed to a unified lockdown throughout the country in 2021. There existed notable heterogeneity in the restrictions imposed on various facets such as closing of schools, sealing country borders, as well as limiting domestic and international flights across the different countries.

**Variability in observed pandemic-related metrics across countries.** Overall for the year 2021, the mean CFR for Pakistan and Nepal were higher than the other three countries, and the mean TPR for India and Pakistan were almost half of that of the others (Fig 7). COVID-19 vaccine doses administered per million people was the highest for Sri Lanka followed by India, while the other countries lagged by a wide margin. For all countries except Nepal, the time-varying reproduction number approximately oscillated between 0.5 and 2. Nepal experienced a sudden surge in $R_t$ to 8 around February 2021 for a few days which eventually stabilized (S2 Fig). There did not appear to be any significant change in mobility in Nepal compared to other countries that could explain this $R_t$ surge in early 2021 (S6 Fig). In 2021 Bangladesh and Pakistan saw a high percentage increase in mobility from baseline for all mobility categories except residential mobility. On the other hand, Sri Lanka saw the highest percentage decrease in mobility from baseline for most of 2021 for the same categories. India and Sri Lanka respectively had the highest reported COVID-19 deaths per million in the first and the second half of 2021 (Fig 7). However, reported death counts can be misleading since countries may underreport deaths to varying extent due to lack of preparedness for the pandemic and the vulnerabilities in healthcare infrastructures. In particular, the IHME estimated URFs of COVID-19 deaths in India, Bangladesh, Pakistan and Sri Lanka were around 7.5, 12.5, 20 and 1 respectively and exhibited minimal temporal variation in 2021 (Fig 7).

**Reported deaths analysis.** Detailed comparisons of the counterfactual death counts (Fig 8 and S4 Fig) and the counterfactual $R_t$ profiles (S2 and S3 Figs) along with the original estimates for each country illustrate how seemingly minor differences in $R_t$ can lead to staggering differences in mortality. For instance, India's observed $R_t$ and counterfactual $R_t$ differed by 0.5 between April and June 2021 when Sri Lanka is the donor but the counterfactual death counts were disproportionately large (4–5 times the observed counts). The differences in pandemic management policies and the extent to which their timing and their effectiveness reduced transmission ($R_t$) are important factors influencing better outcomes in our counterfactual scenarios. Therefore we analyse and compare the pandemic performances of counterfactual scenarios **in terms of reported and total deaths**.

The cumulative number of counterfactual deaths per million on June 30, 2021 based on reported data from the first half of 2021 indicate India's mortality would be much lower if it mimicked pandemic-related policies and population behavior of Pakistan or Nepal and implemented them at the same stage of its pandemic (Fig 9). Our analysis shows that, mortality in

## Reported and Total Deaths Per Million

**Fig 7. Daily reported and total (Reported + Unreported) deaths per million in 2021 for each country.** The red and the black lines correspond to the daily observed and total death trajectories respectively.

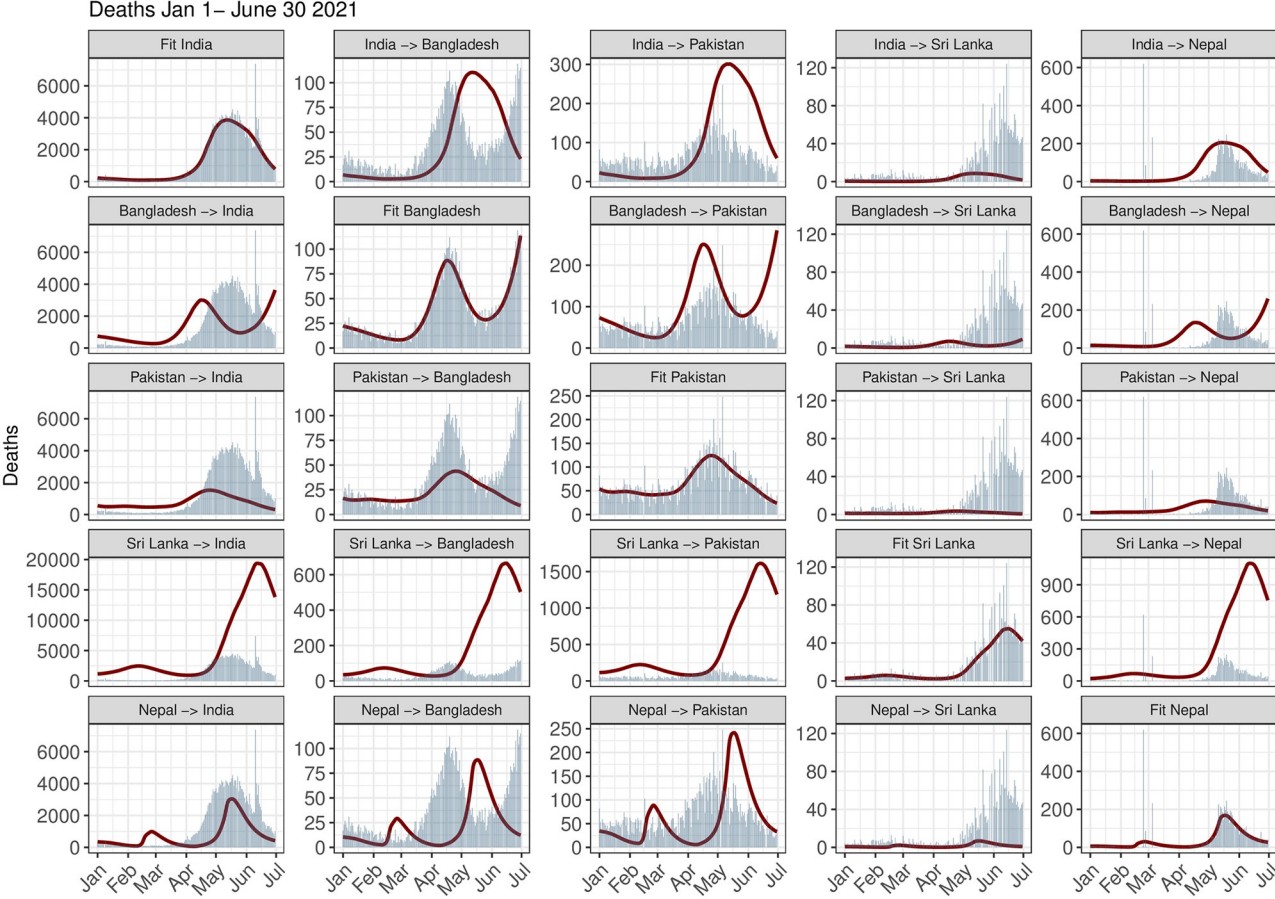

**Fig 8. Daily reported deaths for the different counterfactual situations along with the actual fitted deaths for each country.** The blue bars in the plots denote the actual daily death cases for the recipient country, while the red lines denote the counterfactual ones. The time period of analysis is from January 1, 2021, to June 30, 2021.

India in terms of reported deaths could have been reduced to 96 and 102 deaths per million compared to actual 170 reported deaths per million had it adopted the policies of Nepal and Pakistan respectively. Bangladesh would not experience any significant change in mortality if it had similar pandemic performance as India or Pakistan but could have averted few deaths per million by following Nepal's policies and behavior. Pakistan's mortality would have worsened if it followed policies or behavior of any country other than Nepal. A similar conclusion can be drawn for Nepal. On the other hand, Sri Lanka could have averted a hundred deaths per million if it adopted policies or behavior of any of the other four countries. In the second half of 2021, however, India and Pakistan led in terms of pandemic performance, followed by Bangladesh, Nepal and Sri Lanka. LMICs other than Pakistan would have lower reported mortality had they followed India's strategy. Although overall Pakistan appears to be more successful in keeping COVID-19-related deaths under control throughout 2021 compared to other countries, the daily counterfactual death counts of Pakistan are not uniformly higher than its observed death counts (Fig 8 and S4 Fig). This is also reflected in temporal variations in $R_t$ where the observed and the counterfactual $R_t$ profiles cross each other multiple times within each half of 2021 (S2 and S3 Figs).

| Donor \ Recipient | India | Bangladesh | Pakistan | Sri Lanka | Nepal |
|---|---|---|---|---|---|
| **January 1 - June 30, 2021** | | | | | |
| India | 170 | Counterfactual: 41[24, 72]; Actual: 41; Averted: 0 [-31, 17] | Counterfactual: 85[55, 165]; Actual: 54; Averted: -31[-111, -1] | Counterfactual: 24[13, 43]; Actual: 125; Averted: 101[82, 112] | Counterfactual: 418[266, 648]; Actual: 220; Averted: -198[-428, -46] |
| Bangladesh | Counterfactual: 165 [101, 270]; Actual: 170; Averted: 5 [-100, 69] | 41 | Counterfactual: 87[48, 151]; Actual: 54; Averted: -33[-97, 6] | Counterfactual: 25[11, 52]; Actual: 125; Averted: 100 [73, 114] | Counterfactual: 367[220, 617]; Actual: 220; Averted: -147[-397, 0] |
| Pakistan | Counterfactual: 102[61, 149]; Actual: 170; Averted: 68[21, 109] | Counterfactual: 25 [15, 45]; Actual: 41; Averted: 16[-4, 26] | 54 | Counterfactual: 15[8, 25]; Actual: 125; Averted: 110[100, 117] | Counterfactual: 201[126, 326]; Actual: 220; Averted: 19[-106, 94] |
| Sri Lanka | Counterfactual: 746[430, 1258]; Actual: 170; Averted: -576[-1088, -260] | Counterfactual: 204 [91, 546]; Actual: 41; Averted: -163[-505, -50] | Counterfactual: 397[213, 881]; Actual: 54; Averted: -343[-827, -159] | 125 | Counterfactual: 1794[890, 2961]; Actual: 220; Averted: -1574[-2741, -670] |
| Nepal | Counterfactual: 96[63, 147]; Actual: 170; Averted: 74[23, 107] | Counterfactual: 23[14, 39]; Actual: 41; Averted: 18[2, 27] | Counterfactual: 50[31, 82]; Actual: 54; Averted: 4[-28, 23] | Counterfactual: 14[7, 28]; Actual: 125; Averted: 111[97, 118] | 220 |
| **July 1 - December 31, 2021** | | | | | |
| India | 48 | Counterfactual: 16[11, 27]; Actual: 79; Averted: 63[52, 68] | Counterfactual: 21[14, 31]; Actual: 29; Averted: 8[-2, 15] | Counterfactual: 3[2, 5]; Actual: 455; Averted: 452[450, 453] | Counterfactual: 6[3, 13]; Actual: 72; Averted: 66[59, 69] |
| Bangladesh | Counterfactual: 239[156, 333]; Actual: 48; Averted: -191[-285, -108] | 79 | Counterfactual: 103[60, 154]; Actual: 29; Averted: -74[-125, -31] | Counterfactual: 11[6, 18]; Actual: 455; Averted: 444[437, 449] | Counterfactual: 18[11, 32]; Actual: 72; Averted: 54[40, 61] |
| Pakistan | Counterfactual: 70[46, 109]; Actual: 48; Averted: -20[-61, 27] | Counterfactual: 22 [14, 44]; Actual: 79; Averted: 57[35, 65] | 29 | Counterfactual: 3[2, 6]; Actual: 455; Averted: 452[449, 453] | Counterfactual: 7[3, 16]; Actual: 72; Averted: 65[56, 69] |
| Sri Lanka | Counterfactual: 1627[1231, 2081]; Actual: 48; Averted: -1579[-2033, -1183] | Counterfactual: 1367 [1009, 1933]; Actual: 79; Averted: -1288[-1854, -930] | Counterfactual: 1459[1099, 1903]; Actual: 29; Averted: -1430[-1874, -1070] | 455 | Counterfactual: 884[494, 1593]; Actual: 72; Averted: -812[-1521, -422] |
| Nepal | Counterfactual: 160[124, 199]; Actual: 48; Averted: -112[-151, -76] | Counterfactual: 138[102, 181]; Actual: 79; Averted: -59[-102, -23] | Counterfactual: 147[103, 195]; Actual: 29; Averted: -118[-166, -74] | Counterfactual: 42[23, 73]; Actual: 455; Averted: 413[382, 432] | 72 |

**Fig 9. Cumulative reported deaths per million with the 95% CrI for the five different countries in each of the counterfactual situations in 2021.** The rows correspond to the countries which are the donors, while the columns correspond to the recipients. The diagonal cells denote the observed deaths per million in the 5 countries. The off diagonal cells correspond to the counterfactuals, which consists of 3 quantities, Counterfactual Deaths Per Million obtained using the Relative Approach, observed Deaths per million, and difference of deaths between the reported and counterfactuals that could have been averted. A cell is coloured red or blue depending upon whether the 95% CrI of the number deaths averted in a counterfactual situation lies entirely in the negative or positive side.

Note, India and Bangladesh have similar pandemic performance in the first half of 2021 (averted deaths 95% $CrI_{Bangladesh \to India}$ [−100, 69], $CrI_{India \to Bangladesh}$ [−31, 17]), Bangladesh and Pakistan also have similar pandemic performances (95% $CrI_{Pakistan \to Bangladesh}$ [−4, 26], $CrI_{Bangladesh \to Pakistan}$ [−97, 6]) yet India appears to benefit from adopting Pakistan's pandemic policies (95% $CrI_{Pakistan \to India}$ [21, 109]) (Fig 9). This is due to the asymmetry in counterfactual death counts of two countries arising from differences in $R_0$, $R_t$ and population sizes between donor and recipient countries. As argued before, relatively small changes in $R_t$ could lead to large differences in number of deaths and cases due to varying population sizes (in particular for the large population size of India).

**Total estimated deaths analysis.** Due to time-varying differences in the extent of under-reporting of COVID-19 deaths across countries, we looked at cumulative reported and esti-mated total counterfactual death counts. In the first half of 2021, Pakistan continues to lead in pandemic performances, Bangladesh exhibits similarity to Pakistan, and Nepal's pandemic

performance is no longer better than others when URFs are accounted for (Fig 10). In terms of total deaths, India could have averted 481 and 466 deaths per million had it adopted the policies of Bangladesh and Pakistan. In the second half, however, the relative positions of the countries in terms of pandemic performances remain unaltered regardless of whether we compare reported deaths or total (reported and unreported) deaths. This is not unexpected given that URFs of COVID-19 deaths stabilized for each country in the second half of 2021 (Fig 3). Throughout 2021, the IHME estimated total deaths for Sri Lanka was almost same as the observed deaths.

## Sensitivity analysis results

**Effect of prior specifications on IFR.**    In this analysis, we explored two different parameter specifications for the normal prior distribution of the Infection Fatality Rate (IFR), specifically denoted as (mean, sd) equal to (0.05, 0.03) and (0.12, 0.06), respectively. The counterfactual results from both of these parameter specifications (S7 and S8 Figs) exhibit precisely the same overall pattern as the original analysis (as illustrated in Fig 8). The only

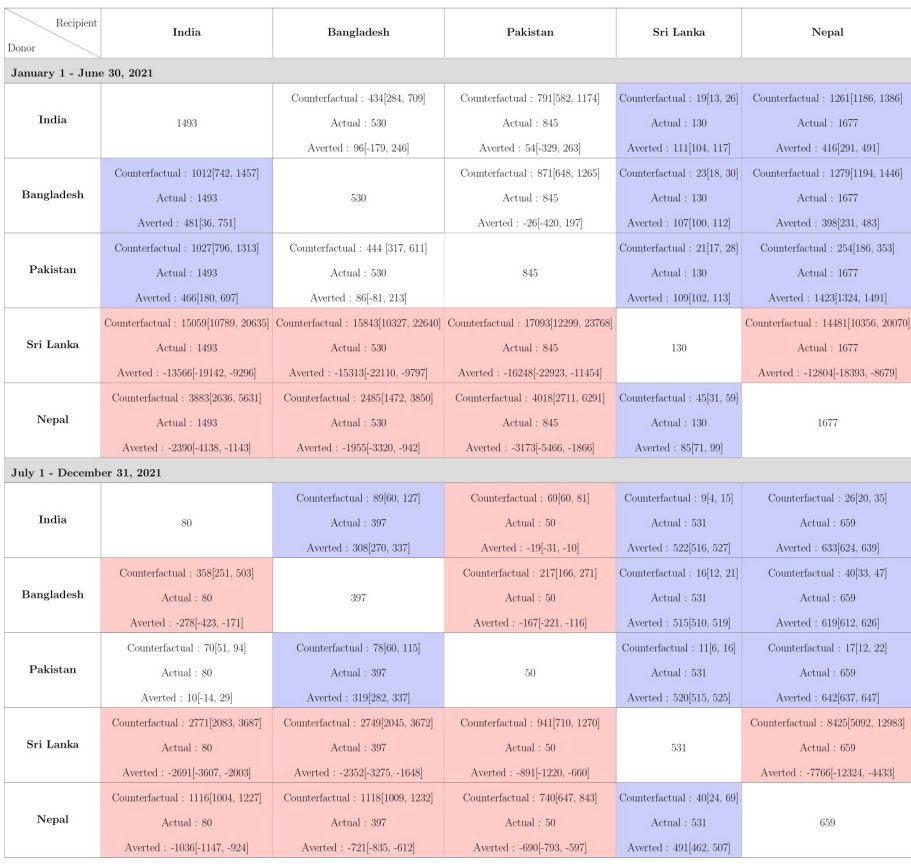

**Fig 10. Total cumulative deaths (reported + unreported) per million with the 95% CrI in 2021 for the five different countries in each of the counterfactual situations.** The rows correspond to the countries which are the donors, while the columns correspond to the recipients. The diagonal cells denote the total deaths per million estimated by IHME in the 5 countries. The off diagonal cells correspond to the counterfactuals, which consists of 3 quantities, Estimated counterfactual Deaths Per Million, estimated Deaths per million by IHME, and difference of deaths between the estimated IHME deaths and counterfactuals that could have been averted. A cell is colored red and blue depending upon whether the number of deaths averted in a country is negative or positive respectively.

difference lies in the magnitude of the peak in the curves, which is inflated in the first specification and deflated in the second. It is important to reiterate that our counterfactual analysis maintains a relative perspective. The primary relative rankings of countries in terms of their pandemic performance have remained consistent throughout this sensitivity analysis even if changes are observed for the actual counts of mortalities in these counterfactual settings. The serosurvey estimates in India were roughly at a six months interval. During the period of 2020 and 2021, four serosurveys were conducted in India. The 95% confidence intervals corresponding to the assumed IFR prior distributions contained the estimates obtained from these serosurveys. Moreover, both the serosurveys in 2021 provided similar IFR estimates. Sharma et al. [33] performed a sensitivity analysis for time varying IFR vs. Fixed IFR where they used a similar Bayesian Semi-Mechanistic Model. They observed that the results using time varying IFR did not deviate significantly from the ones obtained using fixed IFR. They obtained the data for time varying IFR in the UK using the office for national statistics (ONS) data. However, such detailed COVID-19 data were unavailable for the five countries of interest.

**Effect of generation time distribution.** Xu et al.(2023) [34] inferred that the mean intrinsic generation times for Alpha, Delta and Omicron variants are 5.86 [5.47–6.26], 5.67 [3.79–7.55] and 6.84 [5.72–8.6] days respectively. Therefore in this analysis we chose the means of the Gamma distribution for generation to be 5, 6 and 7 days respectively with sd 4.2 which covers the generation times for all the three variants of COVID. The counterfactual outcomes exhibited minimal variation upon adjusting the mean generation time, as depicted in S9–S11 Figs. The alterations primarily manifested as slight adjustments in the actual mortality counts. Notably, the relative ranking of the five countries concerning their pandemic performances remained consistent with the original analysis.

**Effect of including mobility as a predictor of $R_{t,m}$.** Due to lack of information on the prior value for the coefficient of mobility, we compared counterfactual results using two different choices of mean values of the assumed prior distributions, specifically $\mathcal{N}(\log(2), 0.1)$ and $\mathcal{N}(\log(1.5), 0.1)$. The counterfactual results from the mobility analysis (S12 and S13 Figs) suggest that the pandemic performances of Pakistan and Bangladesh were inferior compared to the findings of the original analysis. Overall from the mobility analysis it appears that Nepal and India implemented the best measures followed by Sri Lanka. Pakistan and Bangladesh could have averted a significant number of deaths due to COVID-19 in all the counterfactual situations where they were the recipients. The counterfactual mortality curves exhibit an identical pattern for both prior mean values of the coefficient of mobility, in the $R_{t,m}$ model with the only distinction being that the peaks of the curves associated with a prior mean value of 2 are greater than those corresponding to a value of 1.5 (S12 and S13 Figs). However, the relative ranking of the countries in terms of pandemic performance remains the same in changing the prior distribution parameters.

S6 Fig illustrates the percentage variations in mobility across six distinct categories, specifically Transit Stations, Retail and Recreation, Grocery and Pharmacy, Parks, Workplaces, and Residential, for the year 2021 within the context of all five countries. Nonetheless, one must exercise caution when drawing conclusions. S6 Fig underscores the significant influence of percent changes in mobility on the counterfactual results derived from this mobility analysis. Notably the countries with higher percent changes in mobility, specifically Bangladesh and Pakistan in this context, exhibited relatively poorer pandemic outcomes compared to other nations. This approach of incorporating mobility into the $R_{t,m}$ model makes it quite driven by the mobility data. There may be other variables like mobility that play a role in determining the $R_{t,m}$ series. Small changes in values of $R_{t,m}$ can lead to large changes in counterfactual death estimates so we should be cautious while using covariates in this model. S3 Table shows there

is a high positive correlation between $R_{t,m}$ estimates with or without mobility. The correlation with the `EpiEstim` estimates (based on COVID-19 reported cases data) is higher for the $R_{t,m}$ estimates from the original analysis without mobility. In light of these considerations, we present the findings derived from the original analysis in the main text and mention the mobility analysis as a part of our sensitivity analysis.

## Discussion

### Why is this work important?

In this paper, we focus on comparing the effect of PHIs on reported and estimated total deaths via a counterfactual assessment framework for five south Asian countries, namely, India, Bangladesh, Pakistan, Sri Lanka, and Nepal in 2020–2021, with the main focus on 2021. While geographically proximal, these countries differ significantly in terms of economic, cultural, religious factors, age pyramid and most importantly, population-level comorbidities and healthcare infrastructure like number of hospital beds as observed from Fig 1 [35]. The considerable similarities and differences in these determinants of pandemic outcomes motivate a comparison of the five countries in terms of their policy implementation and efficacy.

Using a renewal process-based Bayesian transmission model from *Mishra et al.* (2021) [6], we adopted a counterfactual-based approach where one country could hypothetically experience the transmission patterns of the pandemic quantified by time-varying reproduction numbers of another country. Using this $R_t$ transfer schema, we analyzed the consequences of shifting policies across the five countries of interest for the two halves of 2021 separately and for 2020. We considered a relative approach for constructing the counterfactuals to preserve certain inherent features of the recipient countries while artificially and hypothetically modifying the policy interventions. Since relatively small changes in $R_t$ can lead to huge variations in the number of cases and deaths, it is not reasonable to compare the countries in terms of their time-varying reproduction numbers only, and we focused on comparing the number of deaths in different countries in different counterfactual situations. To this end, the counterfactuals offer an improved common ground to compare the mortality situations over absolute comparisons of the death time series. Since data capture and coverage of death registration systems differ across the five countries, using the estimated total deaths as opposed to reported deaths add to the novelty of this work.

### What are the major findings and key takeaways?

In 2020 in terms of reported deaths, except for Nepal, in all the other counterfactual scenarios where India is the recipient, India could have averted a significant number of deaths. Overall, our study in 2020 in terms of reported mortality, highlights that Bangladesh and Pakistan made slightly better decisions than the other countries, while Nepal performed the worst in terms of pandemic evaluation. Sri Lanka had a negligible number of reported deaths in 2020. In context of total estimated (reported and unreported) mortality in 2020, Bangladesh, Pakistan, and India demonstrate similar trends in pandemic control, while Nepal appears to fare the worst. Therefore the pandemic performance of India in terms of total deaths is better compared to reported deaths in 2020.

On the other hand, our analyses for the reported deaths in 2021 indicate that Pakistan and Nepal undertook the overall best measures on the whole of 2021 followed by Bangladesh. India lagged slightly in the first half of 2021, but starting from July 2021, it recovered substantially, whereas Sri Lanka performed the worst among the five countries through 2021. The counterfactual analyses for total deaths indicate that Pakistan and India led throughout, followed by Bangladesh, while Nepal lagged behind. Like in case of the reported deaths, Sri Lanka was

behind the other countries in terms of total deaths too. Evaluation of the under reporting factor for cases for all the five countries at the end of 2021, using the results from the model fitting and the serosurveys in India shed further light on the data quality differences across the countries.

## What are the major limitations of this work?

**The quality of input reported deaths**. As mentioned in the Introduction, the five countries of interest faced a lot of challenges in regard to the quality of the reported deaths data. The major possible factors owing to this challenge are high population density, limited testing capacity, variation in reporting across different regions, government transparency and inadequate death registration. These factors contribute towards high under reporting of mortality due to COVID-19 [36–40].

**The quality of input excess death estimates**. First, the reliability of COVID-19 excess mortality estimates, in particular, the individual country-specific estimates have been questioned by several authors [41–46] who caution us against their use in policy evaluations. As a comparative tool we include a table (S7 Table) presenting the disparities in excess mortality estimates along with 95% Confidence Interval using the three most prominent and comprehensive global COVID-19 mortality estimation studies: IHME, WHO [12], and The Economist [13] for both 2020 and 2021.

One critique directed at IHME is around the apparent implausibility of their estimates for certain countries such as Japan or Italy. Namely, the estimates for these two countries suggest that, without COVID-19, average expected mortality rates would have declined, which contradicts historical trends. Another noteworthy criticism of IHME pertains to its significant overestimations of deaths for certain regions, particularly sub-Saharan countries like Kenya and European countries like Denmark, when compared to the WHO estimates. S7 Table reveals that the IHME's combined estimate of excess mortality for India in 2020 and 2021 is the lowest among all three methods. Moreover, both the Economist and IHME estimates for these five countries demonstrate similarity when compared to WHO estimates. This gives us more confidence in going forward with the IHME predictions.

It is also crucial to emphasize that the primary focus of the analysis presented in our paper is on the **relative ordering of the five countries** in terms of pandemic trajectory performance rather than the absolute magnitude of the actual death counts. In terms of combined excess mortality in 2020 and 2021 together, the IHME's relative ordering (from highest to lowest excess deaths per million attributed to COVID-19) is: Nepal > Pakistan > India > Bangladesh > Sri Lanka. In contrast, the Economist's relative ordering is India > Bangladesh > Nepal > Pakistan > Sri Lanka. Lastly, the WHO's relative ordering is India > Nepal > Pakistan > Bangladesh > Sri Lanka. The apparent discordance between the rankings (i.e., that none of the three methods agree with each other in terms of the relative ordering of the five countries regarding excess mortality estimates) shows the general lack of consensus and potential difference disparities in their assessments. Consequently, we agree that it is essential for researchers and policymakers to exercise caution and be mindful of the limitations and uncertainties when utilizing estimates from any of these three methods, not just limited to IHME.

**The choice of prior on IFRs**. Second, while there is existing knowledge on the infection fatality rates for India using the serosurveys, for the other countries, the IFRs are calculated using an additional assumption: the ratio of case fatality rates is the same as the ratio of infection fatality rates. Although this is not a completely invalid assumption to make, more granular information on these metrics from the other countries would have been of help. Moreover due to unavailability of the serosurvey estimates in India on a regular basis, we adopted a fixed

prior for Infection Fatality Rate (IFR) to ensure the identifiability of the parameter estimation in our models within a year of analysis (different for 2020 and 2021).

**Consideration of lives versus livelihoods**. Moreover, the lives and corresponding livelihoods lost from COVID-19 bore a compounding cost on the economy. *Zimmermann et al.* (2021) [8] found that this cost amounted to at least 30% (based on reported and unreported COVID-19 deaths) of India's GDP, when applying the value of a statistical life estimate by *Majumder & Madheswaran,(2018)* [47] to the meta-analyzed mortality estimates. *Karunathilaka* (2021) [48] studies the positive and negative impacts of COVID-19 with reference to challenges on the supply chain in South Asian countries. *Deyshappriya*(2020) [49] reveals that there was a greater potential of increasing poverty incidence related to all poverty types under COVID-19 pandemic situation in Sri Lanka and *Ali et al.* (2020) [39] studies the effects of COVID-19 in Pakistan from a sociological and economic perspective.

Finally, one major limitation of the fundamental premise of this work is that it may be entirely infeasible to impose the restrictions mandated by one country to another. For a diverse country like India, it is almost impossible to impose an uniform set of rules and regulations all over the country, which has been pointed out in recent literature [50]. The National lockdown in 2020 on 1.4 billion people will remain historic and possibly unprecedented. A more sensible way of swapping the intervention scenarios between countries would be to map spatial (administrative units) and temporal (pandemic timeline and waves) on a case-by-case basis to ensure improved calibration of the real scenarios as opposed to a fictional national level counterfactual setting we considered.

## What are some open problems that merit future investigation?

**Incorporating uncertainty in excess death estimates**. Using estimated total deaths as input in the model does not account for the huge uncertainty in these estimates. There is no way to validate the model against ground truth as unreported deaths are and will remain unobservable. A way to mitigate this would be to construct synthetic pandemic timelines based on known generative epidemiological models and construct and evaluate the counterfactuals based on the synthetic data. We leave these options for future exploration in a more integrated hierarchical framework.

**Modifying the approach for generation of counterfactuals**. We generated the counterfactual scenarios by interchanging the time-varying reproduction number ($R_t$) series for each country. This framework assumes that $R_t$ serves as a representative measure of the COVID pandemic for a country as a whole, capturing the effects of innate immunity and other factors unique to each country. Incorporating this $R_t$ transfer schema in our analysis implicitly assumes that the complex interplay of various factors that influence the pandemic's trajectory in each country is summed into this one metric. Other studies such as the one conducted by *Salvatore et al.* [14] have also employed $R_t$ to generate counterfactual situations. However, generating counterfactuals in a meaningful way is a potential vast area of research.

There are alternative ways to model the differences in countries based on varying factors like innate immunity, subclinical infections, vaccinations, population based serological surveys, patterns of mobility and several others. For example, there are cogent arguments that besides different population pyramids, both innate immunity and sub-clinical infections might have played a role in lower levels of COVID mortality in South Asia [51]. *Shyamsunder et al.* [52] study the impact of vaccination on COVID-19 outbreaks by categorizing infected people into non-vaccinated, first dose-vaccinated, and second dose-vaccinated groups and exploring the transmission dynamics of the disease outbreaks. There exist other COVID models that directly integrate serosurvey estimates into the $R_t$ modeling, while others treat

serosurvey data as a distinct compartment within the model, as demonstrated in the studies by *Quick et al.* [53] and *Zhou et al.* [54], respectively. There have been recent developments in implementing synthetic control methods for evaluating the effectiveness of different interventions on the spread of COVID-19 contagion [55–57].

In summary, despite the limitations in the formulation of methods adopted in this study, it is important to recognize the data quality differences while evaluating policies in the global south. With better data and transparent methods we can arrive at less biased comparisons across countries in terms of their pandemic performance. Naïvely comparing the reported number of total deaths at a national level across countries, without normalizing by age structure, population size and differences in death reporting, the ultimate objective of data-driven policy making will always remain elusive to a policymaker.

## Supporting information

**S1 Text. Infection modeling.** The details of modeling true number of infections due to COVID-19.
(PDF)

**S1 Table. Important metrics for the Covid-19 pandemic in the five countries in 2020.** CFR, $R_0$ and TPR denote Case Fatality Rate, Basic Reproduction Number and Test Positive Rate respectively. For CFR and TPR, mean, min, and max of the daily CFR or TPR in the period March 15, 2020–December 31, 2020 reported. For cases and deaths, total counts per million reported on December 31, 2020.
(TIF)

**S2 Table. Important metrics for the COVID-19 pandemic in the five countries in 2021.** CFR, $R_0$, and TPR denote Case Fatality Rate, Time Varying Reproduction Number, and Test Positive Rate respectively. For CFR (or TPR), mean, minimum, and maximum of the daily CFR (or TPR) during the period January 1, 2021–December 31, 2021, are reported. Cumulative number of vaccines per million during the period January 1, 2021 to December 31, 2021 are reported. For cases (or deaths), the cumulative counts per million are obtained using the cumulative number of cases (or deaths) for the whole of 2021.
(TIF)

**S3 Table. Comparison between different $R_{t,m}$ estimates for all the countries in the first half of 2021 (Jan 1, 2021–June 30, 2021).** Estimated $R_{t,m}$ series are obtained using three ways: Not including mobility (original analysis- only based on reported mortality data), including mobility into the model and finally using the R Package *EpiEstim* which calculates $R_{t,m}$ based on COVID-19 reported cases data. *EpiEstim* using the method "parametric_si" and specifying the mean and sd (4 days and 1 respectively) for the Delta strain of COVID-19.
(PNG)

**S4 Table. Different data sources used in our analysis along with relevant website links.**
(PNG)

**S5 Table. Table for Under Reporting Factors (URF) along with their 95% CI for Cases over the one year period starting from March 15, 2020 to Dec 31, 2020 for the different countries.** For each country, we report the URF for cases on June 1, September 1 and December 31.
(PNG)

**S6 Table. Table for Under Reporting Factors (URF) along with their 95% CI for Cases over the one year period starting from Jan 1, 2021 to Dec 31, 2021 for the different countries.**

For each country, we report the URF for cases on April 1, July 1, Oct 1 and December 31.
(PNG)

**S7 Table. Total mortality estimates per million (reported and unreported) in 2020, 2021 and combined total till 2021 by IHME, WHO and the Economist for the five countries of interest.**
(PNG)

**S8 Table. Timeline of COVID-19 interventions in India, Bangladesh, Nepal, Pakistan, and Sri Lanka from March 15-December 31, 2020.**
(PDF)

**S9 Table. Timeline of COVID-19 interventions in India, Bangladesh, Nepal, Pakistan, and Sri Lanka from January 1-December 31, 2021.**
(PDF)

**S1 Fig. Time-varying reproduction numbers for the different counterfactual situations for each country during March 15, 2020-December 31, 2020.** The black lines in the plots denote the actual $R_t$ for the recipient country, while the red lines denote the counterfactual ones. This plot is from the analysis using reported deaths data.
(TIF)

**S2 Fig. Time-varying reproduction numbers for the different counterfactual situations for each country during January 1, 2021-June 30, 2021.** The black lines in the plots denote the actual $R_t$ for the recipient country, while the red lines denote the counterfactual ones. This plot is from the analysis using reported deaths data.
(TIF)

**S3 Fig. Time-varying reproduction numbers for the different counterfactual situations for each country during July 1–December 31, 2021.** The black lines in the plots denote the actual $R_t$ for the recipient country, while the red lines denote the counterfactual ones. This plot is from the analysis using reported deaths data.
(TIF)

**S4 Fig. Daily reported deaths for the different counterfactual situations along with the actual fitted deaths for each country in second half of 2021.** Recipient countries vary along the columns while the donor countries vary along the rows. The blue bars in the plots denote the actual daily death cases for the recipient country, while the red lines denote the counterfactual ones. The time period of analysis is from July 1, 2021 to December 31, 2021.
(TIF)

**S5 Fig. Daily percent change from baseline for different mobilities in 2020.** Baseline is the median value for the corresponding day of the week during January 3, 2020–February 6, 2020. Percent changes are reported for Transit Station, Retail and Recreation, Grocery and Pharmacy, Parks, Workplaces, and Residential for the year 2020 (March 15, 2020–December 31,2020). Google Mobility data used for this plot.
(TIF)

**S6 Fig. Daily percent change from baseline for different mobilities in 2021.** Baseline is the median value for the corresponding day of the week during January 3, 2020–February 6, 2020. Percent changes are reported for Transit Station, Retail and Recreation, Grocery and Pharmacy, Parks, Workplaces, and Residential for the year 2021. Google Mobility data used for this

plot.
(TIF)

**S7 Fig. Daily reported deaths for the different counterfactual situations along with the actual fitted deaths for each country in the first half of 2021 where the (mean,sd) for the IFR prior is given by (0.05,0.03).** Recipient countries vary along the columns while the donor countries vary along the rows. The blue bars in the plots denote the actual daily death cases for the recipient country, while the red lines denote the counterfactual ones. The time period of analysis is from Jan 1, 2021 to June 30, 2021.
(PNG)

**S8 Fig. Daily reported deaths for the different counterfactual situations along with the actual fitted deaths for each country in the first half of 2021 where the (mean,sd) for the IFR prior is given by (0.12,0.06).** Recipient countries vary along the columns while the donor countries vary along the rows. The blue bars in the plots denote the actual daily death cases for the recipient country, while the red lines denote the counterfactual ones. The time period of analysis is from Jan 1, 2021 to June 30, 2021.
(PNG)

**S9 Fig. Daily reported deaths for the different counterfactual situations along with the actual fitted deaths for each country in the first half of 2021 where the mean for the generation distribution is 5 days.** Recipient countries vary along the columns while the donor countries vary along the rows. The blue bars in the plots denote the actual daily death cases for the recipient country, while the red lines denote the counterfactual ones. The time period of analysis is from Jan 1, 2021 to June 30, 2021.
(PDF)

**S10 Fig. Daily reported deaths for the different counterfactual situations along with the actual fitted deaths for each country in the first half of 2021 where the mean for the generation distribution is 6 days.** Recipient countries vary along the columns while the donor countries vary along the rows. The blue bars in the plots denote the actual daily death cases for the recipient country, while the red lines denote the counterfactual ones. The time period of analysis is from Jan 1, 2021 to June 30, 2021.
(PNG)

**S11 Fig. Daily reported deaths for the different counterfactual situations along with the actual fitted deaths for each country in the first half of 2021 where the mean for the generation distribution is 7 days.** Recipient countries vary along the columns while the donor countries vary along the rows. The blue bars in the plots denote the actual daily death cases for the recipient country, while the red lines denote the counterfactual ones. The time period of analysis is from Jan 1, 2021 to June 30, 2021.
(PNG)

**S12 Fig. Daily reported deaths for the different counterfactual situations along with the actual fitted deaths for each country in the first half of 2021 using mobility data in the transmission (Prior Mean = 1.5).** Recipient countries vary along the columns while the donor countries vary along the rows. The blue bars in the plots denote the actual daily death cases for the recipient country, while the red lines denote the counterfactual ones. The time period of analysis is from Jan 1, 2021 to June 30, 2021.
(PNG)

**S13 Fig. Daily reported deaths for the different counterfactual situations along with the actual fitted deaths for each country in the first half of 2021 using mobility data in the transmission (Prior Mean = 2).** Recipient countries vary along the columns while the donor countries vary along the rows. The blue bars in the plots denote the actual daily death cases for the recipient country, while the red lines denote the counterfactual ones. The time period of analysis is from Jan 1, 2021 to June 30, 2021.
(PNG)

## Author Contributions

**Conceptualization:** Ritoban Kundu, Bhramar Mukherjee.

**Data curation:** Ritoban Kundu.

**Formal analysis:** Ritoban Kundu.

**Funding acquisition:** Bhramar Mukherjee.

**Investigation:** Ritoban Kundu.

**Methodology:** Ritoban Kundu, Swapnil Mishra, Bhramar Mukherjee.

**Project administration:** Bhramar Mukherjee.

**Resources:** Lauren Zimmermann, Bhramar Mukherjee.

**Software:** Ritoban Kundu, Swapnil Mishra.

**Supervision:** Bhramar Mukherjee.

**Validation:** Ritoban Kundu.

**Visualization:** Ritoban Kundu, Debashree Ray, Bhramar Mukherjee.

**Writing – original draft:** Ritoban Kundu, Jyotishka Datta, Debashree Ray, Swapnil Mishra, Rupam Bhattacharyya, Lauren Zimmermann, Bhramar Mukherjee.

**Writing – review & editing:** Ritoban Kundu, Jyotishka Datta, Debashree Ray, Swapnil Mishra, Rupam Bhattacharyya, Bhramar Mukherjee.

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
