## [Decision Letter · Decision Letter 0]

14 Jul 2023

PGPH-D-23-00912

Assessing Effects of Interventions on COVID-19 Mortality in South Asian Countries using Counterfactual-based Transmission Models

Dear Dr. Kundu,

Thank you for submitting your manuscript to PLOS Global Public Health. We have now received reports from 2 reviewers and, after careful consideration, we feel that it has merit but does not fully meet PLOS Global Public Health’s publication criteria as it currently stands. Therefore, we invite you to submit a major revision of the manuscript that addresses each of the points raised during the review process.

As demonstrated in the reports included below, the reviewers highlight important concerns regarding the clarity of your rationale, analysis, and discussion. These concerns, in our view, limit the strength of the study. Therefore we ask you to address them through additional work. Without substantial revisions, it is unlikely we will send the paper back for further review.

We look forward to receiving your revised manuscript.

Kind regards,

Vijaykrishna Dhanasekaran, PhD

Academic Editor

Journal Requirements:

Additional Editor Comments (if provided):

Reviewers' comments:

Reviewer's Responses to Questions

**Comments to the Author**

1. Does this manuscript meet PLOS Global Public Health’s publication criteria? Is the manuscript technically sound, and do the data support the conclusions? The manuscript must describe methodologically and ethically rigorous research with conclusions that are appropriately drawn based on the data presented.

Reviewer #1: Partly

Reviewer #2: Yes

2. Has the statistical analysis been performed appropriately and rigorously?

Reviewer #1: I don't know

Reviewer #2: Yes

3. Have the authors made all data underlying the findings in their manuscript fully available (please refer to the Data Availability Statement at the start of the manuscript PDF file)?

Reviewer #1: Yes

Reviewer #2: Yes

4. Is the manuscript presented in an intelligible fashion and written in standard English?

Reviewer #1: Yes

Reviewer #2: Yes

5. Review Comments to the Author

Reviewer #1: This as an analysis of COVID-19 mortality trends in South Asia for the period 2021 in relation to policies and largely relies upon the IHME excess mortality analysis. It does not strangely provide any data for the 2020 period where much of the virus spread and stringent policies were implemented, and the argument that this was not done given lack of data from Sri Lanka, also does not wash.

My major concern is that the paper does not present any counterarguments as to why the IHME estimates might have been erroneous especially as they did not take into account several local studies assessing mortality estimates from graveyard surveillance and existing DHIS systems. While some impact on mortality and reduction in health system coverage in 2020 continuing to 2021 is undisputed (UNICEF ROSA South Asia study 2021), and recent analysis by Subu Subramaniam from the NFHS-5 rounds in India suggests a small impact on neonatal mortality, a counter analysis using local data on COVID-19 related deaths from India, Pakistan, Nepal and Bangladesh would have greatly strengthened the arguments.

What about the virus spread and development of immunity at population level in 2020 and early 2021, even prior to widespread immunization strategies. There are cogent arguments that besides different population pyramids, both innate immunity and sub-clinical infections might have played a role in lower levels of COVID mortality in South Asia (see BMJ 2021 Jun 25;373:n1544. doi: 10.1136/bmj.n1544.).

The impact of vaccination is considered as a uniform variable whereas vastly different vaccines were used and coverage rates in many countries remained low (and mainly consisted of Chinese vaccines). How was vaccine efficacy factored in? What about several population based serological studies indicating high population level immunity by end 2020/early 2021. Were these

Finally and perhaps most importantly, the policy analysis appears to be largely based on reported events or decisions, as opposed to implementation and is way past the most stringent periods of control (in 2020). By mid 2021 in many places, especially rural populations, it was largely game over for major COVID measures beyond masking. The excess mortality effect of impact of economic downturn in the region, isn't really discussed in the context of mitigation measures and if they were effective or otherwise.

In summary, though laudable, this effort falls short of the level of objective scrutiny any modeling exercise would undergo, and does not sufficiently take into account counterfactuals or arguments that the IHME excess mortality estimates might be gross over-estimates.

Reviewer #2: Quantifying and assessing the impact of interventions on COVID-19 mortality is important to inform the public health management and could help for such other directly transmitted infectious diseases. While measure of mortality usually estimated retrospectively, the real-time changes in the health capacity, settings and interventions (including effective contact tracing) could impact both the denominator and numerators of the measures. Therefore, such study should be carefully adopted for the implementation further.

Authors have used their existing framework to analyse the data from different locations of some low and middle income countries in south Asia. The main concern is the rationale of the study is not clearly presented in the introduction with potential hypothesis, this is not merely the same analysis under their reported framework for UK and European countries. Though several such studies on assessing the direct and indirect impact of COVID-19 related interventions found to be significant for COVID-19 infection, transmissibility, severity and mortality. I found this study is interesting for the art of linking the measures of transmissibility to mortality for assessing impact of these interventions on COVID-19 mortality, but has several scopes of improvements as states below.

1. As mentioned above, authors have used their existing framework to analyse the data from other locations, I found the introduction should be improved by including the rationale of this study to take up clearly.

- I see authors have mentioned the challenges and limitations of the public health setting and data access for these locations and tried to illustrate by providing a table 1 in the introduction itself but the text is not in order to follow for general readers.

- Further several parts should not be in the introduction, “… A detailed description of the different interventions implemented by the five South Asian countries of interest in 2021 are provided in Supplementary Table Interventions. Upon observation of the five countries under scrutiny, it is noteworthy that the commencement of their respective vaccination programs occurred in close temporal proximity. Specifically, the initiation of the vaccination campaigns in these countries took place approximately between the late January and early February in the year of 2021. With the exception of India, all four countries undertook comprehensive nationwide lockdowns for multiple days during the initial half of 2021. There were disparities in the length and commencement of these lockdowns across the countries. Conversely, India implemented region-specific lockdowns within individual states as opposed to a unified lockdown throughout the country in 2021. There existed notable heterogeneity in the restrictions imposed on various facets such as schools, country borders, as well as domestic and international flights across the different countries. …”. This section should be in data section and some parts may be in discussion, I wonder if the authors are using these results and data to establish their hypothesis, that should be clearly mentioned in the text and presented accordingly. In that case, I’d suggest for presenting the evidence to establish the hypothesis, authors could present some summery statistics with the references instead of using Tables directly, which are any way based on secondary data.

2. The supplementary tables are not numbered, rather presented with the ‘table title’ in the text, I am not sure is the requirement of the journal. In general, we numbered them as supplementary table and cite them in the main text as required.

3. The data description is not clear for each level of data and countries. I suggest authors could mention country-wise data diversity and may consider some text from introduction to the data section in case.

4. Authors opted the mobility baselines, calculated as the median value for the corresponding day of the week during January 3, 2020 - February 6, 2020. What is the rationale to consider such base line? Better to mention it clearly in the text with evidence of supports. How this baseline for one month could represent the other timing of the year? By January 2020, the human mobility behavior already started changing even though the Governments had not implemented any specific intervention and travel restrictions. Authors could consider the baseline for each week of a year from the data during earlier years. I have not seen any sensitivity analysis on these baselines over the counterfactual results and hence the final outcomes.

5. In equations 1 and 2, authors used the IFR for reported and total retrospectively and assumed to be constant. While the expected deaths d_t,m was temporally evaluated by introducing the temporal components as daily infection and their density to death from the time of infections as the backward looking.

- First how density (Pi) could be constant across the epidemics, in fact several factors could change the distribution over time? If authors assumed this constant should be clearly mentioned in the text and discuss in the limitation section.

- Authors have rightly considered the unobserved cases in the formulation, but I could not find how they have estimated the unobserved cases over time and not clear in the text. Please clarify or provide the details.

6. The selling part is the ‘linking Rt to the mortality’. I found this part is not clear enough.

- Authors used the renewal process to estimate Rt from daily number of cases. Where generation time distribution is approximated with the serial interval distribution as g ∼ Gamma (6.5, 0.62), but not cited the evidence for this parametric distribution. I wonder authors have evaluated from the line-list data for each country. Please clarify or revise the text with the relevant information.

- Considering the Rt as random process with exponential form is very crude assumption, in fact, the Rt usually considered to follow gamma distribution, which surely an exponential family of distribution but should not be an extreme member of the family. I suggest to provide the evidence for this assumption and cite the original study for this.

- The measure Rt is mostly driven by the susceptibility, but while reconstruction via. random process, the authors estimated e^α0,m, which is R0,m the basic reproduction number for country m. Then multiplied a quantity e^Iϵw(t),m, which fixed to 1 at the start of the epidemic and then follow the random walk. I am not sure how it will ensure to have decreasing trend accounting the depletion of susceptible in the population.

- Finally, the liking of transmissibility to the mortality is not found in the text, I can’t assess this part as it not available in their original article clearly. Please provide the details how these Rt could be translated to the mortality in temporal scale.

7. How the pharmaceutical intervention (vaccination) was used to model along with non- pharmaceutical interventions, the text is not clear to locate the formulation. Please provide the details in the main text itself. Even I couldn’t find then in the supplementary except the data.

8. Accordingly, the result section should be revised, I don’t know why the result section’s title is “Results in 2021”? I found the discussion part is poorly written with no proper literatures’ links and comparison. Several recent publications were not discussed in context. The discussion ended up with no data driven conclusion.

9. I suggest to revise the title for general readers, no need to mention ‘counterfactual-based’ terms in the title itself. In fact, it is not ‘counterfactual-based’ transmission model rather the transmission models were used to analyzed for counterfactual scenarios. Further the terms ‘Assessing Effects’ can be revised as ‘Impact assessment’.

10. In Abstract: “… were contributed by five low and middle income countries (LMIC) countries in the Global South.” Delete extra ‘countries’ word from the sentence.

I suggest authors to improve the manuscript considering the above mentioned points to have the manuscript for the readers of the Journal.

6. PLOS authors have the option to publish the peer review history of their article (what does this mean?). If published, this will include your full peer review and any attached files.

**Do you want your identity to be public for this peer review?** For information about this choice, including consent withdrawal, please see our Privacy Policy.

Reviewer #1: **Yes: **Zulfiqar A Bhutta

Reviewer #2: No

---

## [Decision Letter · Decision Letter 1]

25 Sep 2023

PGPH-D-23-00912R1

Comparative impact assessment of COVID-19 policy interventions in five South Asian countries using reported and estimated unreported death counts during 2020-2021

Dear Dr. Kundu,

Thank you for submitting your manuscript to PLOS Global Public Health. After careful consideration, we feel that it has merit but does not fully meet PLOS Global Public Health’s publication criteria as it currently stands. Therefore, we invite you to submit a revised version of the manuscript that addresses the points raised during the review process.

While both reviewers acknowledge the considerable improvement in your manuscript, Reviewer 2 has raised several significant concerns that require further attention:

Reviewer 2 has noted that the article's flow and language structure have indeed improved. However, they have also observed that the methodological weaknesses identified in the previous review have not been adequately addressed. The responses provided by the authors to address certain major issues, specifically those outlined in Reviewer 2's comments 4, 5, and 6, appear insufficient in resolving the concerns raised. It was anticipated that the authors would conduct additional analyses to bolster their methods and findings, including sensitivity analyses that could have been executed relatively quickly. Unfortunately, these crucial analyses have not been carried out in the revised manuscript.

We look forward to receiving your revised manuscript.

Kind regards,

Vijaykrishna Dhanasekaran, PhD

Academic Editor

Journal Requirements:

1. We ask that a manuscript source file is provided at Revision. Please upload your manuscript file as a .doc, .docx, .rtf or .tex. 2. We have noticed that you have uploaded Supporting Information files, but you have not included a list of legends. Please add 
a full list of legends for your Supporting Information files after the references list.

Additional Editor Comments (if provided):

Reviewers' comments:

Reviewer's Responses to Questions

**Comments to the Author**

1. If the authors have adequately addressed your comments raised in a previous round of review and you feel that this manuscript is now acceptable for publication, you may indicate that here to bypass the “Comments to the Author” section, enter your conflict of interest statement in the “Confidential to Editor” section, and submit your "Accept" recommendation.

Reviewer #1: All comments have been addressed

Reviewer #2: (No Response)

2. Does this manuscript meet PLOS Global Public Health’s publication criteria? Is the manuscript technically sound, and do the data support the conclusions? The manuscript must describe methodologically and ethically rigorous research with conclusions that are appropriately drawn based on the data presented.

Reviewer #1: Yes

Reviewer #2: Partly

3. Has the statistical analysis been performed appropriately and rigorously?

Reviewer #1: Yes

Reviewer #2: N/A

4. Have the authors made all data underlying the findings in their manuscript fully available (please refer to the Data Availability Statement at the start of the manuscript PDF file)?

Reviewer #1: Yes

Reviewer #2: No

5. Is the manuscript presented in an intelligible fashion and written in standard English?

Reviewer #1: Yes

Reviewer #2: No

6. Review Comments to the Author

Reviewer #1: I think that the authors have substantively considered most of my comments and explained things and arguments reasonably. The manuscript stands much improved

Reviewer #2: None

7. PLOS authors have the option to publish the peer review history of their article (what does this mean?). If published, this will include your full peer review and any attached files.

**Do you want your identity to be public for this peer review?** For information about this choice, including consent withdrawal, please see our Privacy Policy.

Reviewer #1: **Yes: **Zulfiqar A Bhutta

Reviewer #2: No

---

## [Editor Report · Decision Letter 2]

10 Nov 2023

Comparative impact assessment of COVID-19 policy interventions in five South Asian countries using reported and estimated unreported death counts during 2020-2021

PGPH-D-23-00912R2

Dear Mr. Kundu,

We are pleased to inform you that your manuscript 'Comparative impact assessment of COVID-19 policy interventions in five South Asian countries using reported and estimated unreported death counts during 2020-2021' has been provisionally accepted for publication in PLOS Global Public Health.

Best regards,

Vijaykrishna Dhanasekaran, PhD

Academic Editor
